# Evolving Approaches to Meet Clinical Hours for Undergraduate Nursing Students during COVID-19

**DOI:** 10.3390/ijerph20115974

**Published:** 2023-05-27

**Authors:** Kimberly Rumsey, Shinu Joy, John Michael Leger

**Affiliations:** School of Nursing, University of Texas Medical Branch, 301 University Blvd., Galveston, TX 77555-1132, USA; kirumsey@utmb.edu (K.R.); jmleger@utmb.edu (J.M.L.)

**Keywords:** COVID-19, clinical hours, nursing education, simulation

## Abstract

Background: The COVID-19 pandemic challenged all healthcare professionals to re-think how patient care is provided. Faculty in nursing schools faced similar challenges in offering adequate clinical hours to nursing students, because of the limited availability of clinical placements. Methods: A faculty in one school of nursing introduced virtual simulation resources to supplement in-person clinical hours. The faculty developed a revised clinical curriculum for students that included weekly objectives and deliverables for virtual simulations. The Simulation Effectiveness Tool-Modified (SET-M) was utilized to evaluate the effectiveness of the virtual simulations. Results: 130 students (88.4%) completed the post-implementation survey. After using the virtual simulations, 50% of the students reported feeling confident in providing interventions that foster patient safety. Furthermore, students reported a sound understanding of disease pathophysiology (60%) and medications (53.8%). The qualitative data indicated that students found the virtual simulations beneficial and a safe learning environment. Conclusion: Pre-pandemic virtual simulations were not used by this school of nursing to replace the traditional in-person clinical experiences. However, the pandemic demonstrated that the utilization of innovative virtual simulations are effective methods for student learning to augment traditional clinical experiences.

## 1. Introduction

The COVID-19 pandemic prompted educators, including those in higher education, to be innovative and flexible in implementing learning strategies for teaching students. In-person classes were quickly moved to online platforms as schools across the United States were shut due to the spread of the virus. Hodges et al. [1] defined emergency remote teaching as “a temporary shift of instructional delivery to an alternate delivery mode due to crisis circumstances,” such as the pandemic. Educators had to quickly identify best practices and adopt new approaches for promoting student engagement and learning through online education. Faculty also had to develop new, innovative teaching strategies utilizing distance or virtual education and prepare coursework for students beyond the traditional paradigms and practices in education [2]. In particular, faculty in nursing schools faced the crucial dilemma on whether to rapidly change the traditional teaching methods used for teaching clinical experiences to undergraduate and graduate nursing students. The development of alternative clinical experiences was necessary to ensure the competencies and performance expectations in the changing environment for clinical education were met. 

The international literature has reported several factors that promote the use of remote learning in healthcare education, as well as obstacles that impede online learning [3,4]. Two beneficial factors identified include the capability for independent study and the ability for playback support during online courses. Playback support allows students to attend the courses when convenient and repeat the content as needed. Furthermore, students like the online learning objectives to be clearly outlined, using quality resources, and that they are readily available [3]. Adequate support and help from the faculty and information technology department were also crucial to student success [5]. Students felt that the severity of their personal COVID-19 situation, including economic issues, had a major impact on their learning [4]. The uncertainty concerning healthcare education and when the university would reopen were also issues [3]. Lastly, it was imperative that the students were self-disciplined and adaptable to the new teaching strategies [4]. Natarajan and Joseph [6] found that students experienced a high level of engagement in the learning, but a decreased social presence and satisfaction level with emergency remote learning.

The major challenge facing educators in schools of nursing was the need to hastily develop and provide innovative and relevant clinical experiences, to maintain student matriculation through their programs and enable timely student graduation [7]. At the beginning of the pandemic, regulatory bodies in the U.S. either changed or waived the required percentage of direct in-person patient care students needed to complete nursing programs successfully. For example, on March 13, 2020, the governor of Texas issues a state Disaster Declaration related to the pandemic [8]. A waiver was granted, allowing students in their final year of a Texas-based school of nursing program to meet clinical learning objectives by exceeding the existing 50% limit on simulated clinical learning experiences [8]. Nursing faculty reacted quickly to this change, resulting in clinical curriculums seeing an increase in the use of simulation, virtual reality, and telehealth [7,9]. Simulation is one of the innovative teaching methodologies that has been used increasingly in nursing schools and clinical staff education that helps students build confidence and prepare for clinical practice [10]. Simulation is able to provide realistic, robust, experiential learning in a safe environment. Low-fidelity manikins have been used for many years for activities such as basic life support and the provision of care, such as personal care and linen changes. Today, a simulation may involve high-fidelity manikins or standardized patients for in-person simulation. Virtual simulation involves partial immersion in a digital learning environment using a computer, tablet, or phone to participate in and experience clinical scenarios. The National Council of State Boards of Nursing (NCSBN) developed the national simulation guidelines for prelicensure nursing programs [11]. These guidelines were designed to guide regulatory bodies in evaluating the readiness of prelicensure nursing programs to use simulation as a substitute for traditional clinical experience. Further, the NCSBN standards assist nursing education programs to establish evidence-based simulation programs for the nursing curriculum.

The purpose of this paper is to describe the process used by an undergraduate faculty, at one school of nursing (SON) in Texas, when faced with transitioning its traditional clinical curriculum to a fully virtual format in response to the COVID-19 pandemic. This paper will further discuss how the faculty evaluated the effectiveness of the adopted virtual clinical tools, as well as additional creative methods the school of nursing continues to implement to meet the mandatory minimum clinical hours requirement, to ensure students receive quality clinical experiences.

## 2. Methods

### 2.1. Standards

Following a review of the current literature, the leaders of the SON’s Medical-Surgical II course identified the International Nursing Association for Clinical Simulation and Learning (INACSL) standards [12] as applicable to the pandemic emergency situation. These evidence-based standards guided the faculty in developing a plan for offering virtual clinical experiences to BSN (Bachelor of Science in Nursing) students, beginning in the summer of 2020. Using the Texas Board of Nursing’s requirements and the INACSL standards for allowing up to 50% simulation activities in a clinical course [13], the faculty designated, following approval from the school’s leadership, a 2:1 hour clinical-to-virtual simulation ratio [14,15]. Utilizing the National League for Nursing’s virtual simulation options [16], the faculty identified which resources would be most suitable as replacement opportunities for in-person clinical hours. Examples of these resources included Lippincott^®^ vSim^®^ for Nursing, ATI^®^ Real Life Scenarios, and Shadow Health^®^.

Lippincott^®^ vSim^®^ for Nursing offers virtual scenarios across various patient populations and healthcare settings, including medical-surgical, gerontology, and pediatric contexts allowing the student to practice and improve their prioritization, clinical reasoning, and decision-making skills [17]. ATI^®^ Real Life Scenarios challenges students to make healthcare decisions that impact patient outcomes in various healthcare settings. ATI^®^ Real Life Scenarios utilizes branch logic creating, a custom pathway based on the students’ chosen intervention, to enhance learning and improve clinical judgment [18]. Shadow Health^®^ is another platform similar to Lippincott^®^ vSim^®^ for Nursing that offers students an opportunity to improve their clinical reasoning through virtual clinical experiences [19]. The review of these products resulted in the selection of Lippincott^®^ vSim^®^ for Nursing and ATI^®^ Real Life Scenarios for virtual clinical learning activities. This work resulted in a modified clinical curriculum based on the content, the required number of clinical hours, and the cohort size of the class (*N* = 147). The SON used this information to determine the number of student licenses to purchase and the number of scenarios the course required. Licenses also needed to be purchased for the faculty (*N* = 13) to facilitate the clinical groups, which included the full-time course faculty and the part-time faculty hired to oversee students in the clinical setting, also known as the “clinical faculty.” 

### 2.2. Course Objectives/Deliverables

The Medical-Surgical II faculty team, led by the designated course coordinator, reviewed the course outcomes and clinical objectives and developed a detailed weekly curriculum for conducting virtual clinical experiences. This outline identified the clinical objectives for the course, the virtual simulation assignment, and how the assignment aligned with the objectives and the expected deliverables for students from each activity. A total of eight weeks were outlined to meet the clinical requirements for the course (See Table 1).

Each clinical session began with a pre-brief meeting, as recommended by the INACSL standards. The pre-brief was facilitated by the clinical faculty to review the expectations and provide students with the virtual scenario to complete. The clinical faculty concluded each clinical day with a post-conference session to provide a debrief on the scenario and review the deliverables to receive clinical credit. In the first week, students were expected to complete the ATI^®^ Real Life chronic obstructive pulmonary disease case study. Students were also required to video record themselves performing cardiac and respiratory assessments to meet the competency checklist requirements. The requirements for the second virtual week included completing the ATI^®^ Real Life kidney disease case study. In addition, during the post-conference session the students needed to demonstrate an appropriate hand-off communication using SBAR (situation, background, assessment, recommendation) and develop a patient teaching handout regarding kidney disease. In the third week, students completed Lippincott^®^ vSim^®^ for Nursing, namely transfusion reaction simulation. Students were expected to develop a written plan of care for the patient, including the nursing diagnosis, goals, interventions, and evaluation. During the fourth week, the students completed Lippincott^®^ vSim^®^ for Nursing on lung cancer. Students were required to identify three priority problems and interventions for each patient problem. The midterm clinical evaluation was completed as well. In the fifth week, students completed Lippincott^®^ vSim^®^ for Nursing on asthma and completed a nursing care plan. In the sixth week, Lippincott^®^ vSim^®^ for Nursing on post-op opioid toxicity was completed, and the care plans submitted from Week 5 were revised based on faculty feedback. The seventh week consisted of completing Lippincott^®^ vSim^®^ for Nursing on the post-op complications scenario and a discussion on the assessment and interventions needed to manage the patient. In the eighth week, the students completed the Lippincott^®^ vSim^®^ for Nursing complex comorbid conditions scenario and were evaluated on their entire clinical performance. Table 1 provides the virtual clinical outline of the course.

Prior to the implementation of the revised clinical curriculum, an orientation was conducted with the course faculty, as none of them were familiar with the use of virtual clinical activities. All the course faculty (*N* = 13) had a minimum of a master’s degree in nursing. To establish consistency across the clinical groups, a one-day virtual orientation was implemented. The orientation included training on Lippincott^®^ vSim^®^ for Nursing and ATI^®^ Real Life Scenarios, the expectations for virtual clinical activities, and how to conduct pre- and post-clinical conferences. The faculty were expected to utilize the virtual clinical scenarios to appropriately prepare and engage students during the virtual clinical activities. The revised course outline was also made available to the students for transparency.

### 2.3. Evaluation

The project utilized a mixed method to evaluate the effectiveness of the virtual tools for clinical experiences, for the summer 2020 cohort of second-semester nursing students. This process utilized a convenience sample. There were no ethical issues with the project. To promote inter-rater reliability and ensure consistency across all clinical groups for the evaluation of students’ performance, all the faculty, including the part-time clinical faculty, participated in active training on how to lead virtual simulations during the one-day orientation session conducted prior to the start of the initiation of the course. The instructional training included how to run the vSim^®^, grading the vSim^®^, and the associated deliverables. To evaluate the efficacy of the virtual clinical project, the faculty identified the simulation effectiveness tool (SET-M) as the preferred method to collect student feedback. The SET-M tool was revised to align with the INACSL standards and the American Association of Colleges of Nursing (AACN) baccalaureate essentials [20] (Leighton et al., 2015). The revised tool was valid and reliable. The overall instrument had an internal consistency (coefficient alpha) of 0.85 [20] (Leighton et al., 2015).

The students completed the SET-M evaluation at the end of the course as part of the course evaluation. The SET-M survey included four subcategories (prebriefing, learning, confidence, and debriefing) comprised of 19 questions that utilize a 3-point Likert scale (strongly agree, somewhat agree, and do not agree) for gathering students’ responses following each exercise. An additional, open-ended short answer-style question was added to the survey to gather students’ objective feedback about the virtual simulation experience, for qualitative data collection. The SET-M survey was incorporated into the student end-of-course evaluation to facilitate the collection of feedback. 

## 3. Results

Of the 147 students in the cohort, more than 88% (*N* = 130) completed the SET-M survey. The SET-M survey evaluated the entire virtual clinical course and tools used. The sociodemographic characteristics of the students were not collected for this process. However, all the students were in their second semester of the four-semester nursing program and had experienced in-person clinical exercises during the first semester courses. In the prebriefing category of the survey, 57.6% (75) of the students strongly agreed and 33.8% (44) somewhat agreed that the prebriefing was beneficial to their learning. Moreover, 50.7% (66) also strongly agreed that the prebriefing increased their confidence in performing virtual simulations. Analysis of the questions from the learning category of SET-M determined that 60% (78) of the students strongly agreed that they developed a sound understanding of disease pathophysiology from the virtual simulation activities. In addition, 53.8% (70) strongly agreed that they developed a sound understanding of medications through this type of learning method. The results from the confidence category demonstrated that 47.6% (62) strongly agreed and 43% (56) somewhat agreed that they felt confident in their ability to prioritize care and interventions for their patients in the simulation exercises. Additionally, 50% (65) strongly agreed, and 42.3% (55) somewhat agreed that they felt confident in providing interventions that foster patient safety. Furthermore, 46.2% (60) strongly agreed, and 43.8% (57) somewhat agreed they were confident in using evidence-based practice in providing patient care. Finally, in the debriefing category, 54.6% (71) strongly agreed that debriefing contributed to their learning and provided opportunities to self-reflect on their performance during the simulation activities. Overall, 56.2% (73) of the students strongly agreed that debriefing allowed them to verbalize their feelings before focusing on the scenario. Additionally, 53% (69) of the students strongly agreed that debriefing provided them with an avenue for constructive evaluation of the simulation. 

The SET-M results highlighted a few key areas in which the virtual simulations were not perceived by students to be as beneficial as an in-person experience. The first was students self-reporting a feeling of a lack of empowerment to make clinical decisions. Only 36.9% (48) strongly felt that the simulations empowered them to make clinical decisions. Second, and surprisingly, just 38.4% (50) of the students strongly agreed that the virtual simulation provided them with an opportunity to practice their clinical decision-making skills. Finally, only 39.2% (51) of the students strongly agreed that they felt a sense of confidence in their assessment skills. Table 2 provides the complete results of the SET-M survey.

A thematic review of the responses to the additional open-ended question, “What else would you like to say about the virtual simulation experience?” revealed mostly positive feedback. One student stated, “It was nice because you have the opportunity to make a mistake and learn from it without harming a real patient. I also liked that you are the nurse and get to make decisions versus being in clinical and mostly just watching.” Another student commented, “The simulation experience allowed me to focus on the disease process and to truly understand what steps I need to take from start to finish to make sure that I am providing proper care for that specific patient.” Some students shared their frustration that the minimum score needed to pass the simulation exercise was difficult to achieve. Additional feedback received included, “vSim can be beneficial in learning the scenario but the minimum grade requirements of an 80 can be difficult to achieve due to the vSim program’s required interventions.” A few students commented that they preferred vSim^®^ simulations more than the ATI real-life scenario simulations. Students commented that although virtual simulations were beneficial, they cannot replace the hands-on experience obtained from in-person clinical activities. One student stated, “Virtual simulation was educational and a good substitute, but in-person clinical and getting the hands-on experience is the best option.” Based on the SET-M results, the virtual clinical tools utilized in this clinical curriculum redesign were effective and worthwhile in providing clinical experiences during the emergent situation.

## 4. Discussion

In congruence with nursing schools worldwide [3,4,21], the transition to the virtual setting proved to be challenging for both the faculty and the students; however, a smooth transition was achieved through both a review of best practices in teaching and maintaining the attention to detail when redesigning the course’s clinical curriculum. The feedback from the students established that virtual simulation as a clinical learning exercise was a viable resource to augment the traditional hands-on, in-person clinical experience. These results are consistent with the existing literature that supports the use of simulation to augment clinical experiences when needed [22,23,24]. Schools of nursing continue to face limited clinical site availability due to staffing shortages and increased student enrollment. In addition, the waiver of the 50% limit restricting the use of simulation hours in lieu of in-person clinical hours [8] has expired. In response, the nursing faculty have continued using a hybrid format of in-person and virtual simulations to meet course clinical requirements.

In addition to vSim^®^, and Real life ATI, the faculty continue to introduce additional creative and non-traditional clinical experiences to offer face-to-face clinical opportunities, when possible. For example, at the height of the COVID-19 pandemic, the faculty identified a county-led COVID-19 vaccination clinic as a unique experience to increase in-person clinical opportunities for students [13]. The students were able to practice intramuscular (IM) injections while helping the community combat COVID-19. In addition, over the past nine months, the course faculty have developed and introduced into the curriculum a standardized patient telehealth simulation that provides students with a different perspective on how to deliver quality patient care. The faculty are also exploring the possibility of expanding the vSim^®^ to an innovative virtual reality platform. 

Two limitations were identified in the transition to virtual clinical activities at the end of the semester during a course team meeting. The first limitation was a lack of evaluation (pre-test and post-test) of students’ knowledge acquisition for each week’s simulation scenario, including pathophysiology, treatment plan, nursing problems, and interventions. Second, due to the urgency of the situation and the complexity of the SON’s procurement process, the faculty were limited in their ability to vet a broad range of resources. 

## 5. Conclusions

This paper outlined the process utilized by one SON’s faculty to urgently transition the in-person clinical curriculum to an online format in response to the pandemic. The virtual platforms chosen (vSim^®^ and ATI^®^ Real Life Scenarios) were beneficial in providing students with clinical experiences in a safe environment. It is important that students consistently meet the educational standards established by their respective states’ boards of nursing while meeting the societal expectations for safe, competent, and skilled care by those in the nursing profession. In light of recent events, including in response to the COVID-19 pandemic and natural disasters, such as hurricanes, blizzards, and uncontrolled fires, faculty in schools of nursing must have some nontraditional strategies (virtual simulations) prepared to respond to these types of emergency situations. It is crucial for faculty to develop collaborative clinical partnerships to ensure optimal experiences are provided to students, ultimately impacting quality patient care, while also recognizing the need for long-term changes in nursing education. States’ boards of nursing must also acknowledge that alternative, innovative learning experiences (vSim^®^ and ATI^®^ Real Life Scenarios) should be examined for suitability as an adjunct to in-person clinical activities. These may help equip student nurses with the clinical competencies necessary to provide safe care as registered nurses.

## Figures and Tables

**Table 1 ijerph-20-05974-t001:** Virtual clinical outline.

Week	Clinical Objectives	Assignments	Deliverables
1	1, 2, 3, 7	Pre-brief: Course expectations ATI case study: COPD (chronic obstructive pulmonary disease) Post-conference: Review physical assessment video	Competency video (10 min) Competency checklist
2	1, 2, 3, 4, 7	Pre-brief: Report (what/where/why/how) ATI real-life case study: Kidney disease Post-conference: Return demonstration on giving report communication/SBAR	SBAR Patient teaching handout
3	1, 2, 3, 4, 5, 6, 7	Pre-brief: Review components of the plan of care V-sim: Lloyd Bennett (transfusion reaction) Post-conference: Discuss plan of care for patient in v-sim, including nursing diagnosis, goals, interventions, and evaluation	Results of v-sim Infection control crossword
4	1, 2, 3, 4, 5, 6, 7	Pre-brief: Virtual day expectations V-sim: Julia Morales and Lucy Grey (lung cancer) Post-conference: Review scenario and deliverables	Results of v-sim Nursing diagnosis list (3 priority and 3 interventions for each) Midterm clinical evaluation
5	1, 2, 3, 4, 5, 6, 7	Pre-brief: Virtual day expectations V-sim: Jennifer Hoffman (asthma) Post-conference: Review scenario and deliverables	Results of v-sim Full care plan: this can be on any of the patients seen in the v-sim
6	1, 2, 3, 4, 5, 6, 7, 8	Pre-brief: Virtual day expectations V-sim: Doris Bowman (post-op opioid toxicity) Post-conference: Review scenario and deliverables	Results of v-sim Care plan revisions if necessary
7	1, 2, 3, 4, 5, 6, 7, 8, 9	Pre-brief: Virtual day expectations V-sim: Vernon Watkins (post-op complications) Post-conference: Review scenario and deliverables	Results of v-sim Care plan revisions if necessary
8	1, 2, 3, 4, 5, 6, 7, 8, 9, 10	Pre-brief: Virtual day expectations V-sim: Sherman “Red” Yoder (complex comorbid conditions) Post-conference: Review scenario and deliverables	Results of v-sim Final clinical evaluation

**Table 2 ijerph-20-05974-t002:** SET-M results.

	Strongly Agree	Somewhat Agree	Do Not Agree
Prebriefing increased my confidence	51%	40%	9%
Prebriefing was beneficial to my learning	58%	34%	8%
I am better prepared to respond to changes in my patient’s condition	46%	43%	11%
I developed a better understanding of the pathophysiology	60%	34%	6%
I am more confident of my assessment skills	39%	42%	18%
I felt empowered to make clinical decisions	37%	44%	19%
I developed a better understanding of medications [Leave blank if no medications]	54%	39%	7%
I had the opportunity to practice my clinical decision making skills	38%	44%	18%
I am more confident in my ability to prioritize care and interventions	48%	43%	9%
I am more confident in communicating with my patient	34%	43%	23%
I am more confident in my ability to teach patients about their illness and interventions	42%	48%	10%
I am more confident in my ability to report information to health care team	41%	47%	12%
I am more confident in providing interventions that foster patient safety	50%	42%	8%
I am more confident in using evidence-based practice to provide care	46%	44%	10%
Debriefing contributed to my learning	55%	38%	8%
Debriefing allowed me to verbalize my feelings before focusing on the scenario	56%	33%	11%
Debriefing was valuable in helping me improve my clinical judgement	52%	38%	10%
Debriefing provided opportunities to self-reflect on my performance during simulation	55%	34%	12%
Debriefing was a constructive evaluation of the simulation	53%	35%	12%

## Data Availability

The data for this project have been provided in Table 2 in its entirety.

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
