# Peer review of "Evolving Approaches to Meet Clinical Hours for Undergraduate Nursing Students during COVID-19"

_ijerph, 2023, doi:10.3390/ijerph20115974_

Round 1
Reviewer 1 Report (New Reviewer)
The manuscript “Evolving Approaches to Meet Clinical Hours for Undergraduate 2
Nursing Students During COVID-19" presents an interesting approach to the collection information of the virtual simulations specifically in covid context.
The “Introduction” section of the manuscript provide extensive revision and with a very good redaction. The review of literature is relevant to their study.
The aim of the study is properly highlighted and justified.
The manuscript, using mixed techniques to evaluate the effectiveness of virtual tools. The presentation of the technique and characterization of the results achieved indicate that the method is quite suitable and in fact could be useful to profundize this aspects in the learning process. However, I would like to receive more information about the data collection:
1. What were the socio-demographic characteristics of the students? were no data collected such as sex, age, grade, etc.?
2. did they sign the informed consent?
3. Was it a convenience sample or what type of sample?
I also miss the description of some scenarios? what were they about?
Regarding the discussion I think it would have to be expanded with other research studies. Currently there are many published studies on simulation, especially in nursing, so I would also expand the literature search.
Further, the manuscript presents a good and actualized bibliography. The study is of interest for the scientific community.
Author Response
Thank you for your feedback. Please see the responses in red below.
However, I would like to receive more information about the data collection:
- What were the socio-demographic characteristics of the students? were no data collected such as sex, age, grade, etc.? This was addressed in lines 189-190.
- Did they sign the informed consent? We added this information in lines 178-179.
- Was it a convenience sample or what type of sample? We added this information in line 166.
- I also miss the description of some scenarios? what were they about? Table 1 addresses the different scenarios used for virtual clinical assignments. For example, V-Sim: Lloyd Bennnet is a transfusion reaction scenario.
- Regarding the discussion I think it would have to be expanded with other research studies. Currently there are many published studies on simulation, especially in nursing, so I would also expand the literature search. Three additional current resources were added to lines 249 -250.
Reviewer 2 Report (New Reviewer)
I found this manuscript very interesting and iti represents a really tangible way to found out solution for the challenging task as it is the clinical training for the undergraduate nursing students.
Author Response
Thank you for your feedback. We appreciate you reviewing our manuscript.
Reviewer 3 Report (New Reviewer)
Evolving Approaches to Meet Clinical Hours for Undergraduate 2 Nursing Students During COVID-19
Thank you for submitting your work, and providing me with the opportunity to review your paper. I hope you find my comments both helpful and constructive as that is my aim.
Abstract
- a comment regarding the abstract, and possibly throughout the paper, is the use of the word Faculty, which you use as both plural and singular, which is not the case, please amend throughout - this is possibly due to the differences in countries on how this word is applied
- 'prepare for their nursing careers' isn't required as doesn't really follow the beginning of this sentence
- Amend the sentence beginning 50% as you should not commence a sentence with a number
- The study only included post questionnaires, how was pre confidence and understanding measured?
- The conclusion needs to add something new to the literature as simulation has been implemented and part of many programmes prior to the pandemic, how was this different
Introduction
- in this context 'trigger' isn't an appropriate academic term
- in the second sentence you state 'shuttered' do you mean shut, or are you referring to something else?
- 'found themselves' is not appropriate academic language
- unsure what is meant by 'clinical encounters'
- how do you redefine competencies, were students learning different competencies, or do you mean redefining how competencies were learnt
- 'minimum performance expectancies' - is this an approach we should be teaching our students, why are we compromising students education? Or have I read this section wrong?
Methods
- it is unclear what you mean by 'pre-conference' and 'clinical faculty' as you have not mentioned a clinical faculty previously as you have only stated faculty
Evaluation
- It is stated 'there were no ethical issues' - what does this mean, did students know you were going to review their responses to questionnaires and present them in a publications? This needs further consideration.
Results
The results are presented logically, they are limited due to the nature of the likert scale applied.
Limitations of your study
- these have not been presented and need to be recognised
Author Response
Thank you for your feedback. You provided a different viewpoint and some thought-provoking comments.
Abstract
- A comment regarding the abstract, and possibly throughout the paper, is the use of the word Faculty, which you use as both plural and singular, which is not the case, please amend throughout - this is possibly due to the differences in countries on how this word is applied. Upon researching the plural versus the singular meaning of the word “Faculty”, we consulted with various colleagues who recommended this website (https://ell.stackexchange.com/questions/63913/faculty-vs-faculties). Based on this information, we did not change the term.
- Prepare for their nursing careers' isn't required as doesn't really follow the beginning of this sentence. The verbiage was removed. Please see line 11
- Amend the sentence beginning 50% as you should not commence a sentence with a number. The sentence has been revised in line 16.
- The study only included post questionnaires, how was pre confidence and understanding measured? Since this project was initiated at the beginning of the pandemic, there was an urgency to quickly offer clinical experiences for students. Therefore, there wasn’t enough time to assess confidence or understanding pre-implementation. The virtual simulations offered to the 2nd-semester undergraduate nursing students were new and something they had not encountered before.
- The conclusion needs to add something new to the literature as simulation has been implemented and part of many programmes prior to the pandemic, how was this different? This has been addressed in lines 20-22.
Introduction
- In this context 'trigger' isn't an appropriate academic term. The word was changed to “prompted” (line 27).
- In the second sentence you state 'shuttered' do you mean shut, or are you referring to something else? Yes, this was corrected in line 30.
- Found themselves' is not appropriate academic language. This sentence was revised in line 37.
- Unsure what is meant by 'clinical encounters'. The word was revised to “experiences” for consistency and clarity in line 39.
- How do you redefine competencies, were students learning different competencies, or do you mean redefining how competencies were learnt. This sentence was revised for clarity in lines 39-41.
- Minimum performance expectancies' - is this an approach we should be teaching our students, why are we compromising students education? Or have I read this section wrong? Deleted the word “minimum” in line 40.
Methods
- It is unclear what you mean by 'pre-conference' and 'clinical faculty' as you have not mentioned a clinical faculty previously as you have only stated faculty. This section was revised to define clinical faculty (lines118-119) and “pre-conference” was changed to pre-brief throughout the manuscript.
Evaluation
- It is stated 'there were no ethical issues' - what does this mean, did students know you were going to review their responses to questionnaires and present them in a publications? This needs further consideration. A voluntary, anonymous course evaluation is distributed to undergraduate nursing students to give them the opportunity to evaluate the course. The SET-M results were extracted from the end of course evaluation. Additionally, the students sign a disclosure at the beginning of the program informing them that their responses and pictures may be used for school purposes. This is why we do not believe there are any ethical issues.
Results
- The results are presented logically, they are limited due to the nature of the likert scale applied. We utilized the SET-M survey as published.
Limitations of your study
- these have not been presented and need to be recognized. Please see lines 267 – 273.
Reviewer 4 Report (New Reviewer)
The manuscript entitled Evolving Approaches to Meet Clinical Hours for Undergraduate Nursing Students During COVID-19 aims to explore students’ viewpoints on using virtual simulations. The topic is exciting and meaningful. However, at the current stage, I cannot recommend it to be published. It should be improved in the following aspects.
Firstly, a simulation effectiveness tool is used to collect students’ viewpoints of virtual simulations. However, the manuscript does not tell readers how the simulation effectiveness tool is designed and the reliability and validity of the instrument. In addition, the simulation effectiveness tool uses a 3-point Likert scale. It is better to evaluate students’ viewpoints with a 5-point Likert scale.
Secondly, it is better to compare the findings in the manuscript with previous studies related to virtual education, which will highlight the significance of the study.
Thirdly, it is better to discuss the question of how to use the virtual simulation resources after COVID-19. As we know, COVID-19 has ceased. If the research finding cannot contribute to education after COVID-19 (I believe it can contribute to education at the current time), the study is meaningless.
Author Response
Thank you for your feedback.
1. Firstly, a simulation effectiveness tool is used to collect students’ viewpoints of virtual simulations. However, the manuscript does not tell readers how the simulation effectiveness tool is designed and the reliability and validity of the instrument. In addition, the simulation effectiveness tool uses a 3-point Likert scale. It is better to evaluate students’ viewpoints with a 5-point Likert scale. The reliability and validity of the tool has been addressed in lines 174-177. The authors chose to use the SET-M tool with the 3-point Likert scale because we followed the best practices as outlined by the INACSL standards.
2. Secondly, it is better to compare the findings in the manuscript with previous studies related to virtual education, which will highlight the significance of the study. Three additional current resources were added to lines 249 -250.
3. Thirdly, it is better to discuss the question of how to use the virtual simulation resources after COVID-19. As we know, COVID-19 has ceased. If the research finding cannot contribute to education after COVID-19 (I believe it can contribute to education at the current time), the study is meaningless. This has been addressed in lines 251 - 256 and 281-284.
Reviewer 5 Report (New Reviewer)
It would be interesting if the authors could present a balance of losses and gains in terms of skills lost and acquired through the use of online platforms.
Although it seems to have had an advantageous result, it is not clear what the future impact of these changes will be, especially on interpersonal relationships.
For the consideration of the authors
Author Response
Thank you for your feedback. You bring up good points for future research such as examining therapeutic communication skills and technical skills for nursing students who primarily receive virtual clinical experiences in their prelicensure program.
Round 2
Reviewer 1 Report (New Reviewer)
I have carefully considered all the comments and suggestions I provided as a reviewer and have made the necessary revisions to the manuscript. I am pleased to report that the authors have adequately addressed all comments raised during the review, which leads me to believe that the study merits publication.
Author Response
Thank you for your feedback. We appreciate you taking the time to review the manuscript.
Reviewer 4 Report (New Reviewer)
The manuscript entitled Evolving Approaches to Meet Clinical Hours for Undergraduate Nursing Students During COVID-19 aims to explore students’ viewpoints on using virtual simulations. The topic is exciting and meaningful. However, it should be improved in the following aspects.
Firstly, please tell readers how the simulation effectiveness tool is designed and the reliability and validity of the instrument in the current study.
Secondly, please redesign the questionnaire with a 5-point Likert scale.
Thirdly, it is necessary to discuss the findings in detail.
Fourthly, it is necessary to discuss how to use the virtual simulation resources after COVID-19. As we know, COVID-19 has ceased. The study is meaningless if the research finding cannot contribute to education after COVID-19 (I believe it can contribute to teaching at the current time).
Author Response
Hello,
Please see the responses in red below.
The manuscript entitled Evolving Approaches to Meet Clinical Hours for Undergraduate Nursing Students During COVID-19 aims to explore students’ viewpoints on using virtual simulations. The topic is exciting and meaningful. However, it should be improved in the following aspects.
Firstly, please tell readers how the simulation effectiveness tool is designed and the reliability and validity of the instrument in the current study.
This was not a research study. It was a quality improvement project. The reliability and validity of the tool were addressed in lines 174-177 with the last revision. We did not evaluate the reliability and validity of the tool in the quality improvement project.
Secondly, please redesign the questionnaire with a 5-point Likert scale.
We cannot redesign the questionnaire used as this project is already complete. The tool was utilized because it was recommended by the International Nursing Association for Clinical Simulation and Learning.
Thirdly, it is necessary to discuss the findings in detail.
We have reported all of our findings/results (quantitative & qualitative) in our manuscript.
Fourthly, it is necessary to discuss how to use the virtual simulation resources after COVID-19. As we know, COVID-19 has ceased. The study is meaningless if the research finding cannot contribute to education after COVID-19 (I believe it can contribute to teaching at the current time).
We address how virtual simulation resources can be used in education post-COVID-19 and it continues to be used in the previous revision. Please see lines 250-256 and 281-284.
This manuscript is a resubmission of an earlier submission. The following is a list of the peer review reports and author responses from that submission.
Round 1
Reviewer 1 Report
Dear Authors thank you so much for sharing your work with us and giving us the privilege to contribute. It is my opinion that the subject studied is very important however it is not clear the following issues:
1. What is the main question addressed by the research?
The aim of this study is to describe the process used by undergraduate faculty at one Texas school of nursing to transition its clinical curriculum to a fully virtual format in response to the COVID-19 pandemic, but what the researchers what to study? The impact/effectiveness of this virtual tools?
2. We consider the topic original and relevant in the field but what does it add to the subject area compared with other published material? The introduction and discussion can be improved regarding the state of the art in this area.
3. We believe that the authors could improve the methodology and specify the type of study, participants, sample, ethics issues, etc
4. The conclusions should be consistent with the evidence and arguments presented and addressed the main question posed.
Reviewer 2 Report
Dear authors:
I think the study makes an important contribution to the audience. It reports an interesting methodology and results for nursing faculty members, in case of pandemic or similar events.
However I recommend some modifications in order to improve the technical excellence of the paper:
- INTRODUCTION: strongly recommend you to add more references in order to show the state of knowledge about the topic from a worldwide perspective.
- RESULTS: lines 108 - 110 and related questions in Table 2: "a better understanding", I am not sure if the students were asked to compare between the virtual simulation activities and the in-person experience or between the virtual simulation activities (VSA) and other theorical or practical activities. This must be clarified. I also think that if this was not clearly explained/clarified to the students, this is a limitation of the study. They should have known if the statement "I developed a better understanding of the pathophysiology" and "I developed a better understanding of medications" are in order to compare the VSA with one thing or another.
- DISCUSSION. There is too much commentary on the findings, but a discussion with the international literature is lacking. If the reason of this is that there is scarce international literature about the topic, then it should be stated. A limitation paragraph is also missing.
- CONCLUSION. I think you should be cautious with the statament in the last sentence (line 183). I do not agree with "alternative, innovative learning experiences can provide student nurses with the clinical competencies necessary to provide safe care as registered nurses"; the Results do not support that.
Reviewer 3 Report
The manuscript entitled "Evolving Approaches to Meet Clinical Hours for Undergraduate Nursing Students During COVID-19" presents the overall student satisfaction of a virtual simulation program in undergraduate nursing students in pandemic.
The use of virtual simulation scenarios was a necessary and indispensable resource during the pandemic and the mobility restrictions imposed worldwide employed in many health sciences faculties. The manuscript evaluated is contextualized in a U.S. state, health sciences undergraduate nursing school, but is not referenced or compared to didactic resources used elsewhere in the world or to the use of similar resources in other undergraduate nursing schools. This deficiency is mostly evident in the introduction and discussion.
The manuscript is brief and interesting, in general terms, but it is incomplete and its consistency is weak, presenting points with important deficiencies to be considered for publication.
The introduction does not contextualize or define the state of the question or justify the need for the study. The objectives are stated ambiguously and are not fully answered in the results. The selection and composition of the sample of teachers and students is not described correctly or sufficiently, nor is the process and its phases.
The methods and instruments used and reflected are appropriate and correctly justified, although it is necessary to expand/include some aspects that clarify the participants and the design.
Although the analyses carried out are adequate and the results are presented in an orderly manner, they do not provide an answer to the proposed objectives The tables presented are adequate and clear, but insufficient. The discussion is not consistent and is not supported by other studies, it is only an exposition of the satisfaction of the authors and the participants (students).
However, the evaluated manuscript is characterized by the absence of essential aspects of conceptualization and justification in the introduction, description of the method, sample and study design, presentation of results and elaboration of the discussion and conclusions.
For these reasons, it is recommended that the present version NOT be accepted for publication.
Specific comments:
1- Introduction:
The wording of the same is concise and in the right direction, from the concrete to the particular.
However, it is recommended that other studies analyzing and describing the use of online teaching resources during the pandemic be included in the state of the question. It is also recommended to describe the structure of the faculty in which the study is conducted, the curriculum. In L52-L54 it is announced that the present study describes the process used by undergraduate faculty at a Texas school of nursing to transition their clinical curriculum to a fully virtual format in response to the COVID-19 pandemic." But this announcement is neither fulfilled nor evidenced in the results,
The terms essential to understanding the manuscript are not described or conceptualized: clinical simulation, virtual simulation, simulation or online...What are the characteristics of each? What are their advantages and disadvantages described in the existing literature?
The problem is justified in the pandemic as a need for methodological change but it is not justified why one methodology or resource is selected and not another...is it necessary to clarify or justify the selection of online simulation as an opportunity, are there previous results in other studies or learning experiences about of previous effectiveness of this methodology?
2. Methods:
2.1 Standards:
The study population (N) is not indicated: neither teachers nor students,
There is no information on the number of teachers involved or their characteristics. There is no description of the sociodemographic characteristics of the teachers of the degree or of the simulation experience. It is necessary to include a description of the teachers involved, their training experience, their experience with clinical simulation and online simulation, and their clinical specialty.
Likewise, it is essential to state the student population and describe the study sample, indicating their sociodemographic characteristics, their course, whether they have previous experience in face-to-face patient care and their experience with clinical simulation, online simulation and virtual simulation.
Point 2.3 refers to the previous training of the teachers, but does not indicate the process, objectives, development or duration of the training.
It is necessary to identify and describe "resources considered appropriate and acceptable as opportunities..." and include the description and specific features of Lippincott® vSim® for Nursing, ATI® Real Life Scenarios, Shadow Health®, etc.
The scenarios used in the virtual simulations and the expected outcomes of each scenario are also not described.
2.2. Course Objectives/Deliverables
Although in lines L82-L85 it is announced that an example of "week 1 of the course outline" with the scenario topic (EPOC) and the competency checklists is presented, it is necessary to describe the remaining weeks and topics, their hourly duration, and their time schedule. It is also appropriate to describe the competency checklists mentioned.
2.3. Evaluation :
The design of the study and the dates on which it is carried out are not indicated, the time frame (month and year) is not precisely determined.
3. Results
Do the SET-M evaluation tool and the results shown in table 2 refer to a specific topic, to a specific simulation, to specific contents, or to the use of the methodological tool used in general terms? It is necessary to clarify whether the results shown are the evaluation of the value of the vSim® application or of a simulation scenario developed through this tool with specific contents?
4. Discussion:
It does not meet the requirements of a discussion, it does not relate, compare or expose in relation to other studies.
5. Conclusion: the conclusions do not include a clear and concrete answer to the proposed objective. It is necessary to focus and give a clear answer to the proposed objectives based on the results obtained.
